# CONTINUAL LEARNING WITH NEURAL ACTIVATION IMPORTANCE

## ABSTRACT

Continual learning is a concept of online learning along with multiple sequential tasks. One of the critical barriers of continual learning is that a network should learn a new task keeping the knowledge of old tasks without access to any data of the old tasks. In this paper, we propose a neuron importance based regularization method for stable continual learning. We propose a comprehensive experimental evaluation framework on existing benchmark data sets to evaluate not just the accuracy of a certain order of continual learning performance also the robustness of the accuracy along with the changes in the order of tasks.

## 1 INTRODUCTION

Continual learning, or sequential learning, is a concept of online learning along multiple sequential tasks. The aim of continual learning is learning a set of related tasks that are observed irregularly or separately online. Therefore each task does not necessarily contain overlapped classes with others and, in worst case, different tasks consist of mutually disjoint classes. Therefore, in continual learning, one of the main challenges is training new tasks with new classes without catastrophic forgetting existing knowledge of prior tasks (and their classes). A model adapts to a new task without access to some or entire classes of past tasks but keeping and maintaining acquired knowledge from new tasks (Thrun, 1996). Since the training of a neural network is influenced more by recently and frequently observed data, a machine learning model forgets what it has learned in prior tasks without continuing access to them in the current task. On the other hand, rigorous methods that maintain the knowledge of entire previous tasks are impractical in adapting new tasks. Thus, researchers have developed diverse methods to achieve both stability (remembering past tasks) and plasticity (adapting new tasks) in continual learning.

There are three major types in previous continual learning methods; 1) architectural approaches modifying the architecture of neural networks (Yoon et al., 2017; Sharif Razavian et al., 2014; Rusu et al., 2016), 2) rehearsal approaches using sampled data from previous tasks (Riemer et al., 2018; Aljundi et al., 2019; Gupta et al., 2020), and 3) regularization approaches freezing significant weights of a model calculating the importance of weights or neurons (Li & Hoiem, 2017; Kirkpatrick et al., 2017; Zenke et al., 2017; Nguyen et al., 2017; Aljundi et al., 2018a;b; Zeno et al., 2018; Ahn et al., 2019; Javed & White, 2019; Jung et al., 2020). Most recent regularization methods have tackled the problem in more fundamental way with regularization approaches that utilize the weights of a given network to the hit. The basic idea of regularization approaches is to constrain essential weights of old tasks not to change. In general, they alleviate catastrophic interference by imposing a penalty on the difference of weights between the past tasks and the current task. The degree of the penalty follows the importance of weights or neurons with respective measurements. Significance of an weight stands for how important the weight is in solving a certain task.

EWC (Kirkpatrick et al., 2017) introduces elastic weight consolidation which estimates parameter importance using the diagonal of the Fisher information matrix equivalent to the second derivative of the loss. However, they compute weights' importance after network training. SI (Zenke et al., 2017) measures the importance of weights in an online manner by calculating each parameter's sensitivity to the loss change while training a task. To be specific, when a certain parameter changes slightly during training batches but its contribution to the loss is high (i.e., rapid change of its gradient), the parameter is considered to be crucial and restricted to be updated while learning future tasks. However, the accuracy of their method shows limited stability even with the same order of tasks of

test data. Unlike Zenke et al. (2017), MAS (Aljundi et al., 2018a) assesses the contribution of each weight to the change of learned function. In other words, it considers the gradient of outputs of a model with a mean square error loss. Gradient itself represents a change of outputs with respect to the weights. The strength of the method lies in the scheme of data utilization. It considers only the degree of change of output values of the network, so that any data (even unlabeled one) is valid to compute the gradient of the learned function with regard to weights. VCL (Nguyen et al., 2017) is a Bayesian neural network based method. It decides weight importance through variational inference. BGD (Zeno et al., 2018) is another Bayesian neural network based apporoach. In this method, it finds posterior parameters(e.g. mean and variance) assuming that posterior and the prior distribution are Gaussian. These methods (Kirkpatrick et al., 2017; Zenke et al., 2017; Aljundi et al., 2018a; Nguyen et al., 2017; Zeno et al., 2018) calculate and assign importance to weights directly as describe in Figure 1a. In order to alleviate interference across multiple tasks, weight importance based approaches such as EWC, SI, and MAS assign importance $\Omega_k$ to each weight $\omega_k$ in the network. However, in the case of convolutional neural networks, weights in the same convolutional filter map should have the same importance. Furthermore, since those methods basically consider the amount of change of weight, it is impossible to reinitialize weights at each training of a new task, which decreases the plasticity of the network. (Additional explanation of weight reinitialization is discussed in section 2.2)

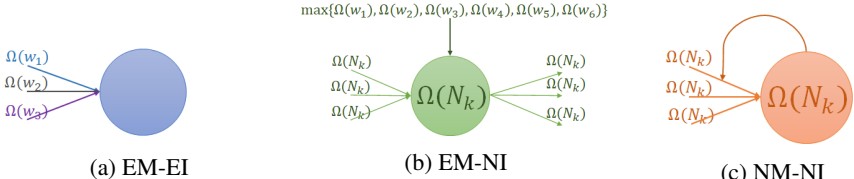

(a) EM-EI  (b) EM-NI  (c) NM-NI

Figure 1: Three different types of Measurement-Importance (a) EM-EI (Edge Measurement, Edge Importance): Weight importance is calculated based on weight measurement. (b) EM-NI (Edge Measurement, Node Importance): Neuron importance is calculated based on weight measurement. Weight importance is redefined as the importance of its connected neuron. (c) NM-NI (Node Measurement, Node Importance): Neuron importance is calculated based on neuron measurement. Weight importance is defined as the importance of its connected neuron.

UCL(Ahn et al., 2019) propose a Bayesian neural network based method to mitigate catastrophic forgetting by incorporating weight uncertainty measurement indicating the variance of weight distribution. They claim that the distribution of essential weights for past tasks has low variance and such stable weights during training a task are regarded as important weight not to forget. As illustrated in Figure 1b, they suggest a node based importance in neural network. First, the smallest variance among the weights incoming to and outgoing from a corresponding neuron decides the importance of the neuron, and then all those weights take the same neurons importance as their weight importance. Ahn et al. (2019) is applicable only to a Bayesian neural network and highly dependent upon hyper-parameters. Furthermore, it is computationally expensive to train compared to Zenke et al. (2017) and our proposed method. Jung et al. (2020) is another recent algorithm based on neuron importance. Its node importance depends on the average activation value. This idea is simple but powerful. Activation value itself is a measurement of neuron importance, and weights connected to the neuron get identical weight importance. This corresponds to the type in Figure 1c.

One of our key observations in prior experimental evaluations is that the accuracy of each task significantly changes when we change the order of tasks, as it is also claimed and discussed in (Yoon et al., 2019) proposing a order robust method of continual learning in their architectural approach. Evaluating with fixed task order does not coincide with the fundamental aim of continual learning where no dedicated order of tasks is given. Figure 2 shows example test results of state of the art continual learning methods compared to our proposed method. Classification accuracy values of prior methods fluctuate as the order of tasks changes(from Figure 2a to Figure 2b). Based on the observation, we propose to evaluate the robustness to the order of tasks in comprehensive manner in which we evaluate the average and standard deviation of classification accuracy with multiple sets of randomly shuffled orders.

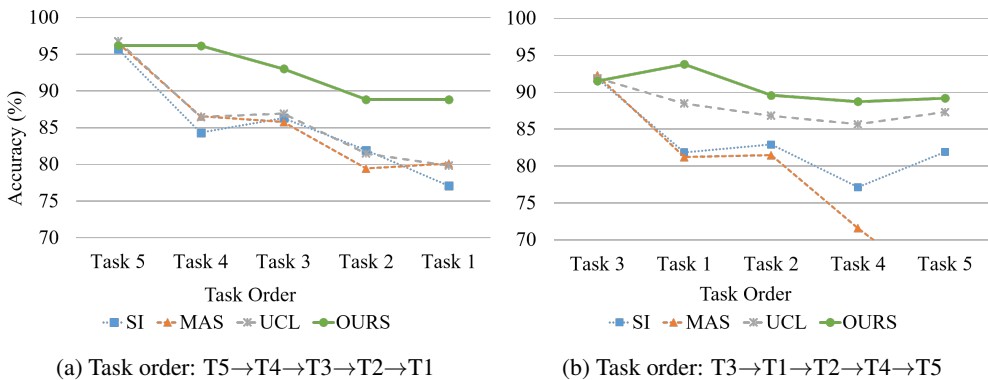

(a) Task order: T5→T4→T3→T2→T1  (b) Task order: T3→T1→T2→T4→T5

Figure 2: Classification accuracy of continual learning on Split Cifar10. SI (Zenke et al., 2017), MAS (Aljundi et al., 2018a) and UCL (Ahn et al., 2019) show critical changes in their performance as the order of tasks changes.

The key contributions of our work are as follows:
• We propose a simple but intuitive and effective continual learning method introducing activation based neuron importance.
• We propose a comprehensive experimental evaluation framework on existing benchmark data sets to evaluate not just the final accuracy of continual learning also the robustness of the accuracy along the changes of the order of tasks. Based on the evaluation framework, existing state-of-the-art methods as well as our proposed method are extensively evaluated.

## 2 PROPOSED METHOD

### 2.1 NEURON IMPORTANCE WITH AVERAGE OF ACTIVATION VALUE

As French (1991) and French (1999) suggested, since activation overlap causes catastrophic forgetting, a successful network is required to build representations with as little overlap of activation as possible across continual tasks, called sparse representations, to mitigate mutual interference. If a network builds dense representation, a small amount of changes in the representation affects the stability of all trained classes since nodes and weights in a network are in casual relations. On the other hand, if a network produces distributed representations for all classes, alteration of representation for certain class does not affect others.

In our method, we adopt NM-NI (Node Measurement, Node Importance) type in Figure 1c based in the measurement of activation value at each neuron. Similar to Jung et al. (2020), we measure the average activation value of $k^{th}$ neuron as its neuron importance $\Omega_k$. Incoming weights connected to the same neurons have the same importance. In the case of convolutional neural networks, the average activation value corresponds to the average activation value of a feature map (i.e., global average pooling value).

Average activation value itself, however, is not able to fully represent the characteristics of an essential neuron. Encoded features at each layer describe different aspects of an input image and, as a result, activation at each layer cannot be evaluated together. Due to the difference in absolute average activation values across the layers, weights of earlier layers tend to be considered more essential as Figure 3 shows. If average activation value is used as neuron importance, method will prefer to keep the weights of earlier layers. Instead, we propose to use layer-wise average activation divided by respective standard deviation for importance measurement. Even though it is a simple extension of (Jung et al., 2020), it prevents a particular layer from getting excessive importance compared to other layers, that, in turn, prevents our network from overfitting to a particular task keeping its plasticity regardless of task order. As Figure 19a shows, for each method, normalized average weight importance of each layer(total 6 layers) is calculated. Prior average activation based regularization term assigns around 57% of total importance to layer 1(57%, 12%, 10%, 6%, 8%, 8%, respectively

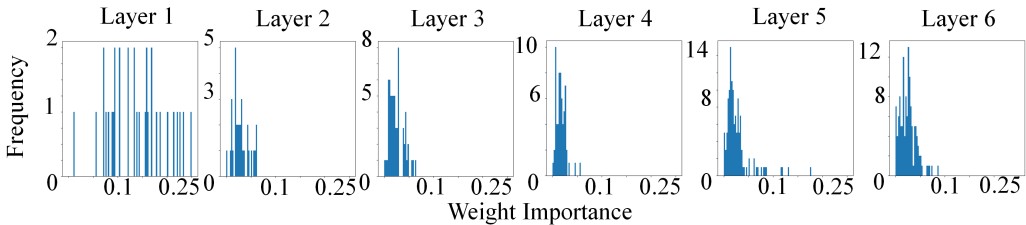

(a) Average Activation Value based weight importance distribution

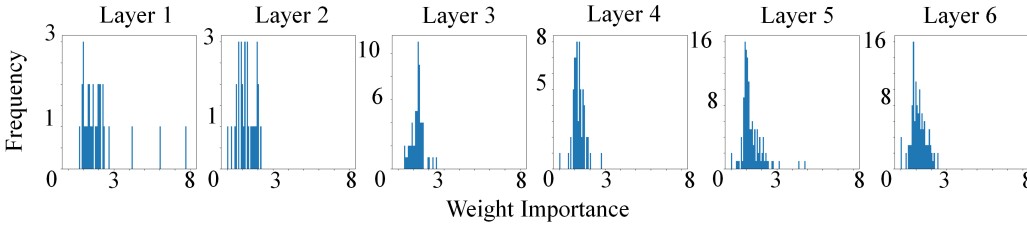

(b) Average Activation / STD based weight importance distribution

Figure 3: Distribution of Weight Importance. Note that range of weight importance value is normalized by dividing the standard deviation value into the average activation value. This is based on the first task of split CIFAR 10.(task order: 2-0-1-3-4)

for the 6 layers). On the other hand, our proposed regularization loss term assigns 26% of total importance to layer 1. Furthermore, our method avoids assigning excessive importance to certain layer(26%, 16%, 16%, 15%, 15%, 12%).

Then, why this improves the continual learning performance regardless of task order? In prior works, more weights of lower layers tend to be frozen in earlier tasks that eliminate the chance of upcoming tasks to build new low-level feature sets. Only a new task that is fortunately able to rebuild higher layer features based on the frozen lower layer weights from previous tasks could survive. On the other hand, ours keeps the balance of frozen weights in all layers securing more freedom of feature descriptions for new tasks in both lower and higher layers. Indeed, lower layer features such as edges are not class(task) dependent features. Therefore, freezing lower layer features actually is not preferable in continual learning. Even though lower level features are mostly changed over tasks, new task may find alternative lower level features keeping its plasticity. On the other hand, if the network fails to freeze some higher level features that are class(task) dependent, it may lose robustness. As we have seen, prior works spend most of their importance to lower layers losing its robustness.

In order to encode such relation, we propose to use average and std. of activation values of each layer rather than using weighs directly. Our loss function is described as follows.

$$L_t = \tilde{L}_t + \alpha \sum_l \Omega_{n_k}^t (w_l^{t-1} - w_l^t)^2, \tag{1}$$

where $\tilde{L}_t$ is loss of current task, $t$ is task index, $l$ is weight index, and $\Omega_{n_k}^t$ indicates $k^{th}$ neuron importance. $\alpha$ is a strength parameter to control the amount of weights consolidation. Neuron importance is defined as follows.

$$\Omega_{n_k}^t = \frac{\frac{1}{N_t} \sum_{i=1}^{N_t} f_{n_k}(x_i^{(t)})}{\sigma + \epsilon}, \sigma = \sqrt{\frac{\sum_{i=1}^{N_t} \{f_k(x_i^{(t)}) - \frac{1}{N_t} \sum_{i=1}^{N_t} f_k(x_i^{(t)})\}^2}{N_t}}, \tag{2}$$

where $N_t$ is the number of instances, $k$ is neuron index, $f_k(\cdot)$ is activation value, and $i$ is instance index. We introduce $\epsilon$ to prevent the numerator from being zero when the standard deviation becomes zero.

## 2.2 WEIGHT RE-INITIALIZATION FOR BETTER PLASTICITY

For continual learning, networks have to not only avoid catastrophic forgetting but also learn new tasks to their fullest. According to the degree of difference in optimal classification feature space of different tasks, optimized feature space in previous task might be significantly changed in a new task. Thus, with a new task, we can let the model start either from random weights or from optimized weights with previous task. Even though the optimized weights with previous task can be considered as one set of random weights for a new task, we hope to avoid any chance of optimized weights for one task working as a local optimal for another similar task that may hinder the training from learning a new global or better optimal feature space. The situation can be explained with $\Omega_k(w_k^{t-1} - w_k^t)^2$ term in the loss function of our network. During a learning of next task, the network is informed of past tasks by $\Omega_k(w_k^{t-1} - w_k^t)^2$ term which lets the network maintain essential weights of past tasks assigning high $\Omega_k$ values. In other words, $\Omega_k(w_k^{t-1} - w_k^t)^2$ delivers the knowledge of previous tasks. Whatever the magnitude of $\Omega_k$ is, however, $\Omega_k(w_k^{t-1} - w_k^t)^2$ term is ignored if $w_k^{t-1}$ almost equals to $w_k t$ already in the initial epoch of the training of a new task, which prevents the network from learning a new task. This situation is alleviated by weight re-initialization that allows the value of $\Omega_k(w_k^{t-1} - w_k^t)^2$ be high enough regardless of the magnitude of $\Omega_k$ in the training of a new task. In this case, still the knowledge of previous tasks will be delivered by $\Omega_k$ and affect the training of a new task. We have verified the effect of reinitialization on Split CIFAR10 data set. We achieve 4% higher average accuracy when we apply weight reinitialization to our algorithm. Moreover, the performance of UCL(Ahn et al., 2019) increases as well with reinitialization. The results are presented in Appendix.

## 3 EXPERIMENTAL EVALUATIONS

We perform experimental evaluations of our method compared to existing state-of-the-art methods for continual learning on several benchmark data sets; split and permuted MNIST (LeCun et al., 1998; Goodfellow et al., 2013), and incrementally learning classes of CIFAR-10 and CIFAR-100 (Krizhevsky et al., 2009). Our algorithm is trained based on the description in Ahn et al. (2019) which has tested existing approaches with different hyper-parameters to find their best performance. We train all different tasks with batch size of 256 and Adam using the same learning rate; 0.001. (Learning rate adaptively changes during learning based on the validation loss.) For the split CIFAR tasks, as aforementioned, we perform the evaluation multiple times shuffling the order of tasks randomly to evaluate the robustness to task orders. "Absolute task order" indicates the sequence of tasks that a model learns. For instance, task 1 stands for the first task that a model learned no matter what classes comprise the task. We test with all 120 combinations of task sequences for split CIFAR10 and 200 random orders for split CIFAR10-100. To minimize statistical fluctuations in accuracy, each combination of task sequences is repeated three times.

We define several evaluation metrics. Task-wise average accuracy (TA accuracy) means the accuracy of each assigned task averaged through the whole continual learning steps. We estimate the degree of forgetting per task via TA accuracy. Learning step-wise average accuracy (LA accuracy) represents the accuracy of each learning step averaged through the whole tasks involved. The degree of interference per learning step is measured by LA accuracy.

### 3.1 MNIST

We first evaluate our algorithm on a split MNIST benchmark. In this experiment, 2 sequential classes compose each task (total 5 tasks). We use multi-headed and multi-layer perceptrons with two hidden layers with 400 ReLU activations. Each task has its own output layer with two outputs and Softmax. We train our network for 40 epochs with $\alpha = 0.0045$.

In Figure 4, we compare task accuracy and LA accuracy. MAS outperforms all other baselines reaching 99.81% while ours achieves 99.7%. However, the accuracy is almost saturated due to the low complexity of data so that the difference of accuracy is not significant. Note that ours greatly outperforms EWC, achieving 99.73% and 98.2% respectively.

We also evaluate methods on Permuted MNIST dataset. Our model used in this evaluation is MLP which consists of two hidden layers with 400 ReLUs each and one output layer with Softmax.

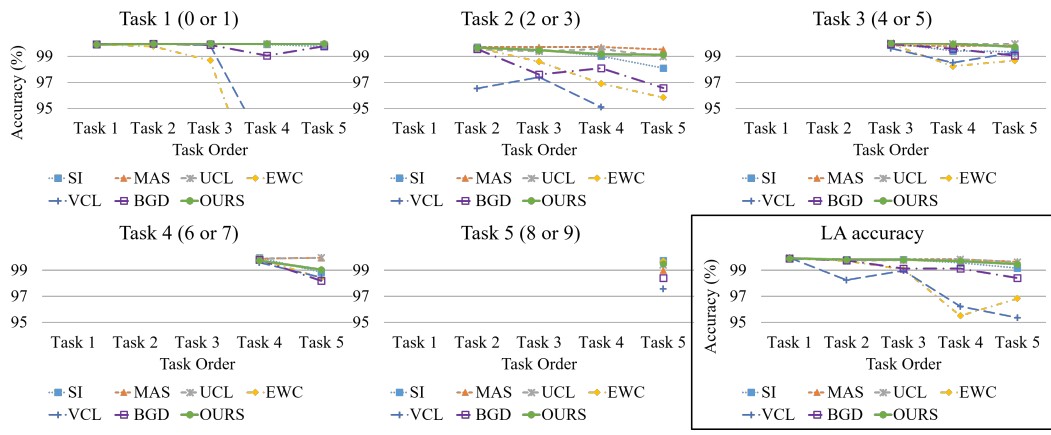

Figure 4: Results on Split MNIST benchmark. Here, VCL indicates VCL(without coreset)(Nguyen et al., 2017).

The network was trained for 20 epochs with $\lambda = 0.005$. Also, to normalize the range of activation value, ReLU is applied to the output layer additionally when computing neuron importance $\Omega_k$. Our algorithm(95.21%) outperforms MAS(94.70%), EWC(82.45%) and VCL(without coreset)(89.76%) and on the other hand, UCL(96.72%), SI(96.39%) and BGD(96.168%) show better results. However most results on this data set achieve almost saturated accuracy.

## 3.2 SPLIT CIFAR10

Evaluation on Split CIFAR10 dataset is based on the multi-headed network with six convolution layers and two fully connected layers where the output layer is different for each task. We train our network for 100 epochs with $\alpha = 0.7$.

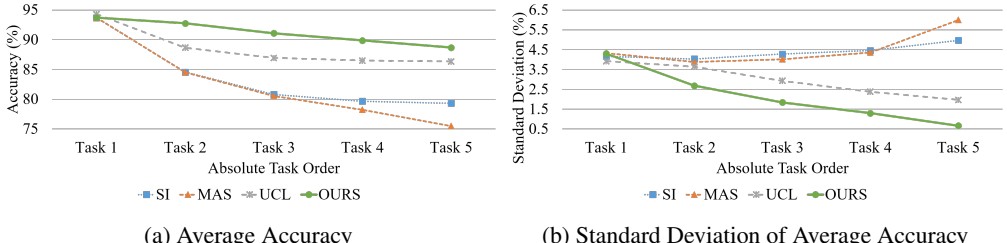

(a) Average Accuracy    (b) Standard Deviation of Average Accuracy

Figure 5: LA accuracy on Split CIFAR10.

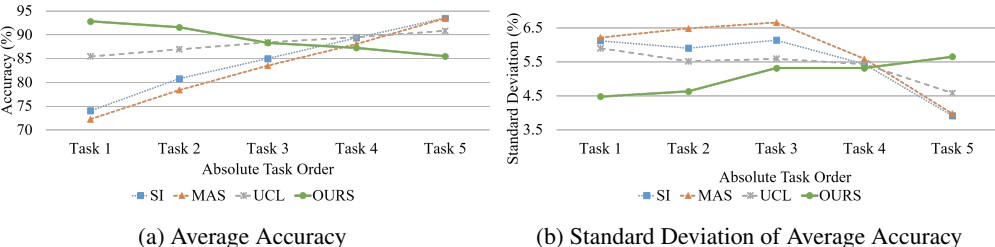

(a) Average Accuracy    (b) Standard Deviation of Average Accuracy

Figure 6: TA accuracy on Split CIFAR10

As Figure 5 shows, our method outperforms all other methods with large margins. Also, 5b shows that our algorithm is more robust to the order of tasks. In Figure 6, we achieve overall highest accuracy than others. Accuracy drop in TA accuracy measurement of our method is natural because

weight importance based methods inevitably increase the number of high importance weights loosing their plasticity gradually. On the contrary, other compared methods show increasing accuracy. It looks like that their plasticities are relatively high, however, it implicitly also indicates that their stabilities are quite limited possibly because weight importance based continual learning is not working appropriately. Note that TA accuracy of task1 in Figure 6a for other methods are relative quite low compared to TA accuracy of task5. Proposed method shows better stability in the order of tasks and also has a low degree of forgetting. In our method, average degraded degree of performance is lowest as $1.23\%$, whereas SI is $18.06\%$, UCL is $7.35\%$, and MAS is $22.89\%$ .

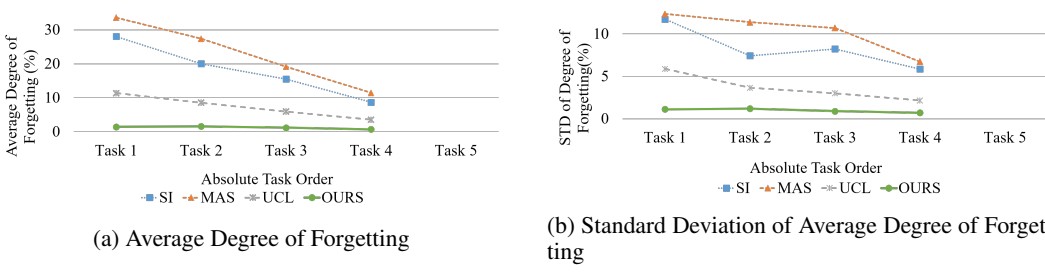

(a) Average Degree of Forgetting

(b) Standard Deviation of Average Degree of Forgetting

Figure 7: Degree of Interference on Split CIFAR10. It is calculated by each task's first learned accuracy - accuracy after learning the last task

## 3.3 SPLIT CIFAR10-100

We evaluate our method on Split CIFAR10/100 benchmark where each task has 10 consecutive classes. We use the same multi-headed setup as in the case of Split CIFAR10. We train our network for 100 epochs with $\alpha = 0.5$. We fix task 1 as CIFAR10 due to the difference in the size of dataset between CIFAR10 and CIFAR100. The order of task 2 to task 11 that consists of CIFAR100 is randomly shuffled. Our algorithm shows better stability showing best accuracy values in old tasks.

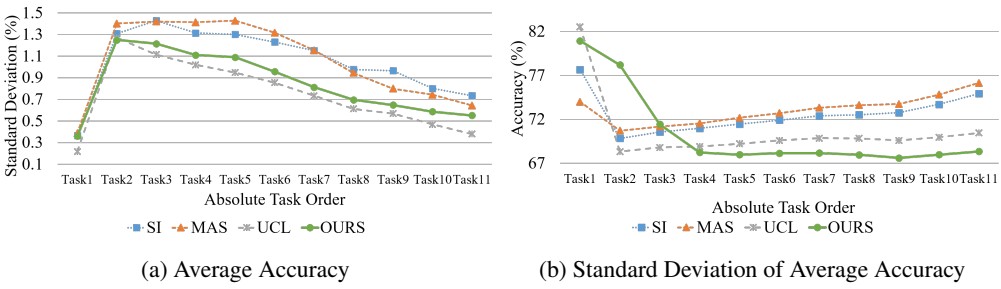

(a) Average Accuracy

(b) Standard Deviation of Average Accuracy

Figure 8: LA accuracy on CIFAR10-100.

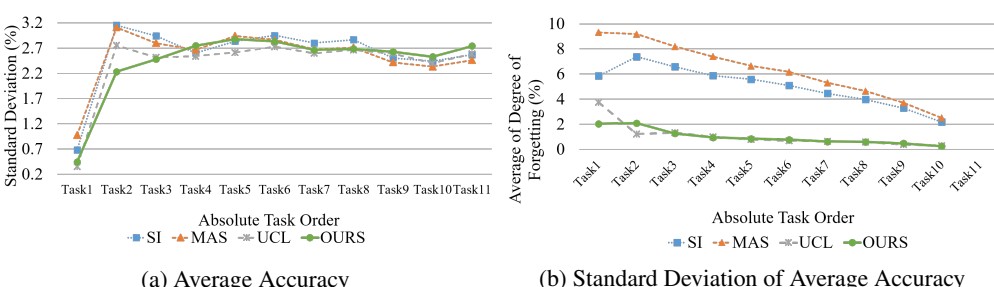

(a) Average Accuracy

(b) Standard Deviation of Average Accuracy

Figure 9: TA accuracy on CIFAR10-100.

On the other hand, previous methods seems to prefer to be better with recent new tasks proving that

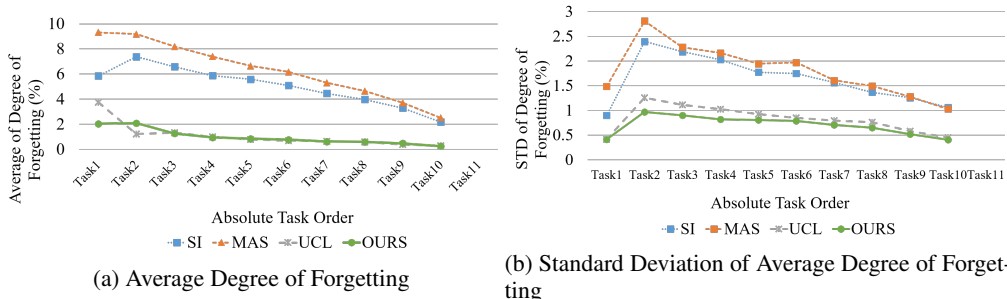

(a) Average Degree of Forgetting

(b) Standard Deviation of Average Degree of Forgetting

Figure 10: Degree of Interference on Split CIFAR10-100. It is calculated by each task's first learned accuracy - accuracy after learning the last task.

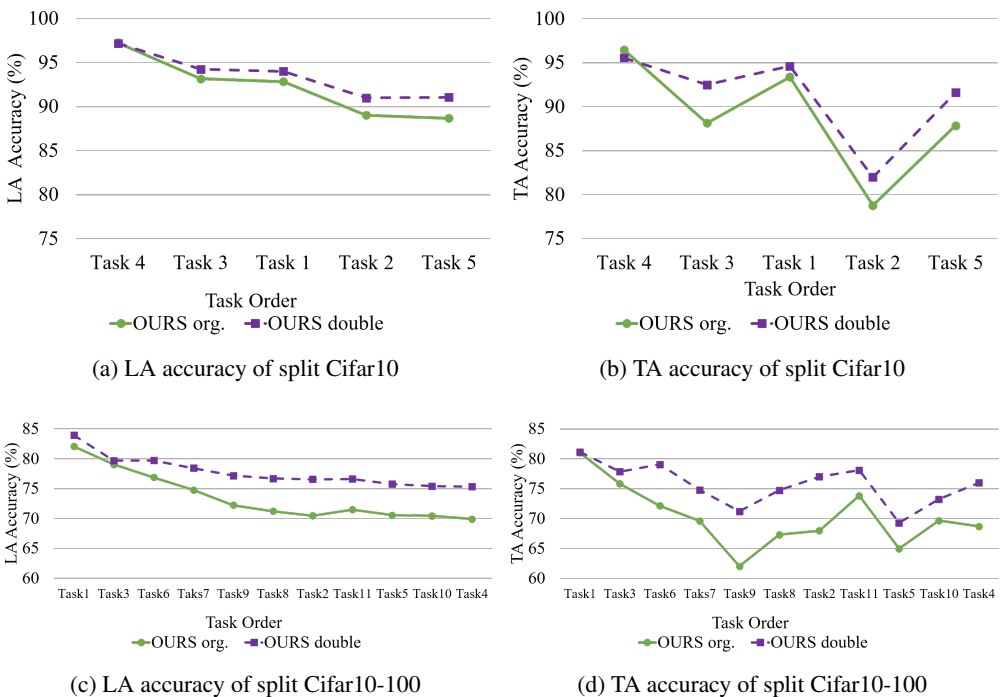

(a) LA accuracy of split Cifar10

(b) TA accuracy of split Cifar10

(c) LA accuracy of split Cifar10-100

(d) TA accuracy of split Cifar10-100

Figure 11: The performance on CIFAR10 and CIFAR10-100 with doubled channel. Accuracy increases when we use a doubled channel network.

our importance based continual learning is working appropriately. Indeed, SI seems that it learns new tasks very well but it forgets what it has learned. In practical, the decrease of plasticity in our method can be addressed by using a larger network. Figure 11 show that our network with doubled number of channels has improved accuracy keeping its stability and better plasticity.

Figure 10 shows that our method obtains lowest average degraded degree of performance among SI, MAS, UCL and ours, achieving $5.02\%$, $6.3\%$, $1.06\%$, and $0.98\%$ respectively. One observation on the stability with CIFAR10-100 data set despite the larger size of it compared to CIFAR10 is that training CIFAR100 after CIFAR10 has the effect of pre-training a network with CIFAR10.

## 4 CONCLUSION

In this work, we have proposed an activation importance based continual learning method. Comprehensive evaluation have proved that the proposed method has implemented importance based continual learning achieving fundamental aim of continual learning tasks balancing between stability and plasticity.

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

# A APPENDIX

## A.1 EVALUATION METHOD

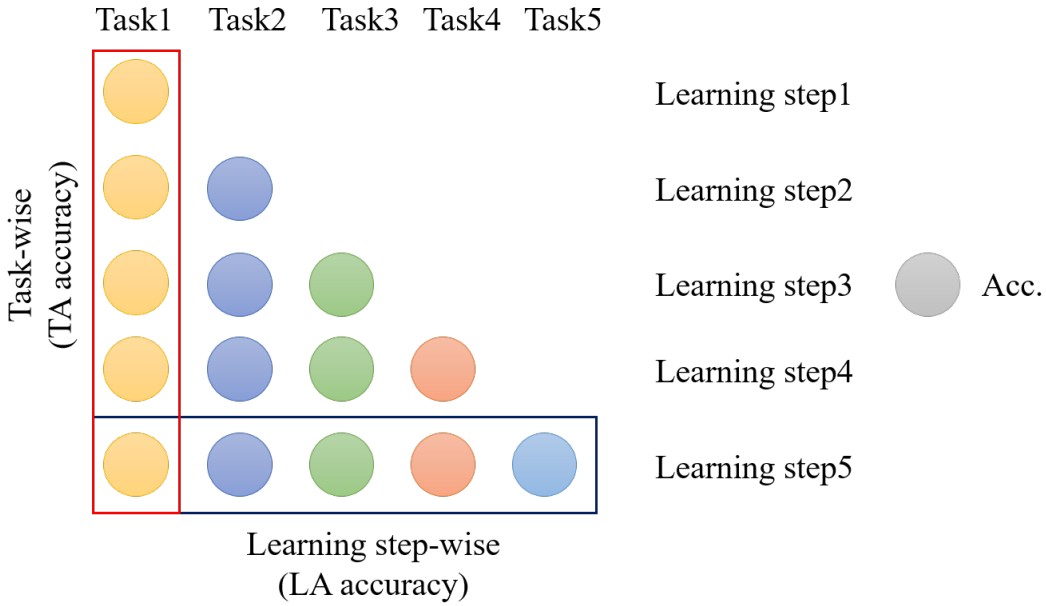

Figure 12: Concept of Evaluation.

## A.2 HYPERPARAMETERS FOR EXPERIMENTS

In MNIST experimnets, we use $\beta = 0.0001$ for UCL, $c = 1.0$ for SI, $\lambda = 4000$ for EWC, $\lambda = 1.0$ for MAS and std_init=0.02 and $\eta = 1$ for BGD(Zeno et al., 2018). In Cifar experiments, we use $c = 0.7$ for SI, $\lambda = 1.0$ for MAS and $\beta = 0.0002$ for UCL.

## A.3 ADDITIONAL ABLATION STUDY OF OUR ALGORITHM

Figure 15 visualizes weight importance values accumulated over tasks. Each elements represents one weight importance and the intensity of color indicates the magnitude of weight importance value. All values are normalized 0 to 1. Each difference map between two consecutive importance maps shows respective weight groups received relatively higher importance values.

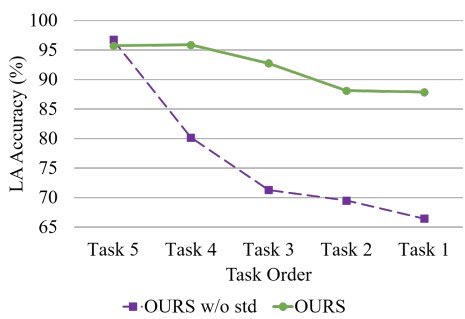

Figure 13: Comparison Performance on Split CIFAR10. Taks Order: T5 → T4 → T3 → T2 → T1. Note that the performance increases when we regard neuron importance as avergae activation value / STD.

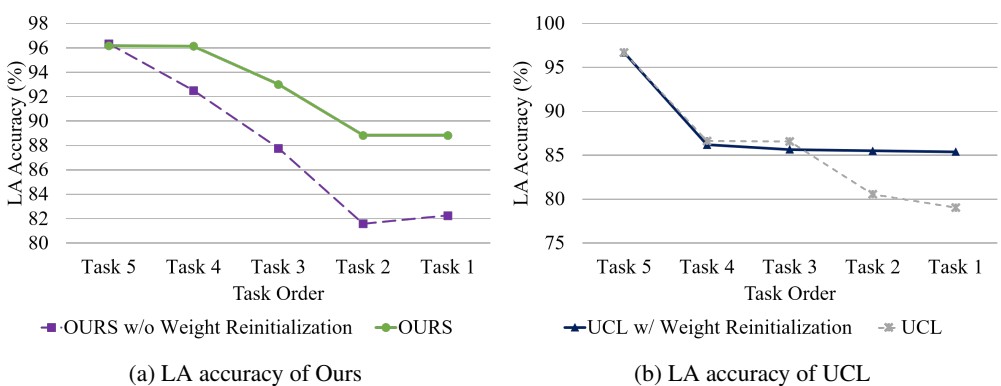

(a) LA accuracy of Ours        (b) LA accuracy of UCL

Figure 14: Comparison Performance on Split CIFAR10 with Ours and UCL. Taks Order: T5 → T4 → T3 → T2 → T1. Note that both performance of Ours and UCL increase when weight re-initialization is applied.

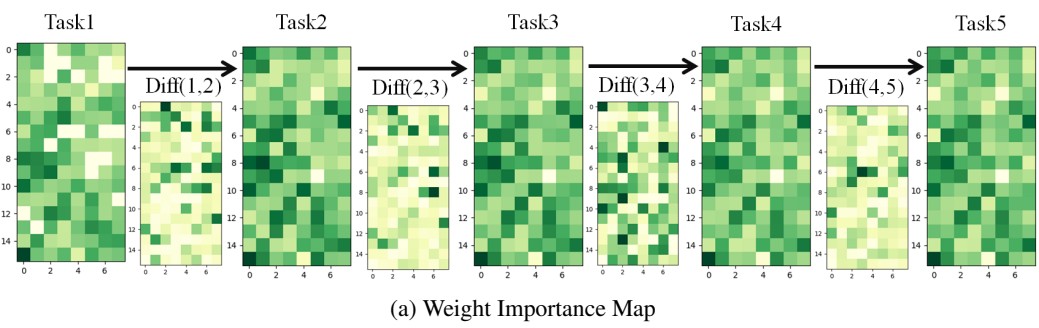

(a) Weight Importance Map

Figure 15: Weight Importance Maps of Continual Tasks

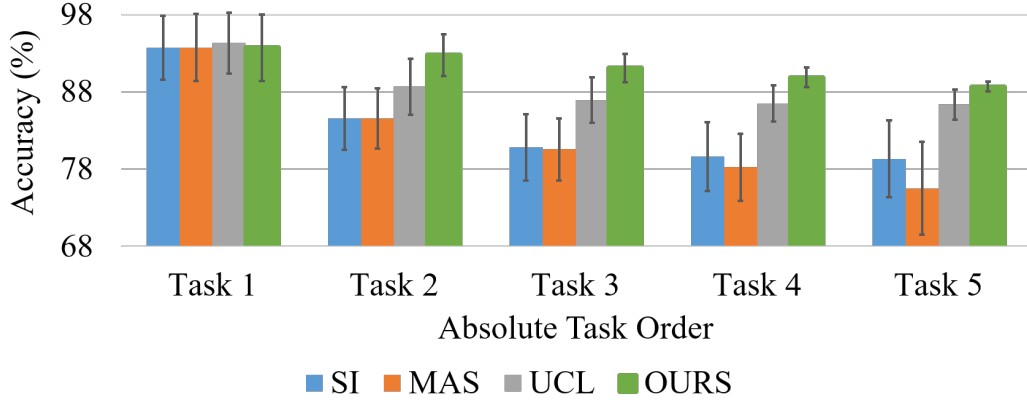

Figure 16: Another representation of split cifar10 LA accuracy. To describe both accuracy and standard deviation at once, we change the form of graph. However, it is difficult to indicate difference of accuracy and standard deviation.

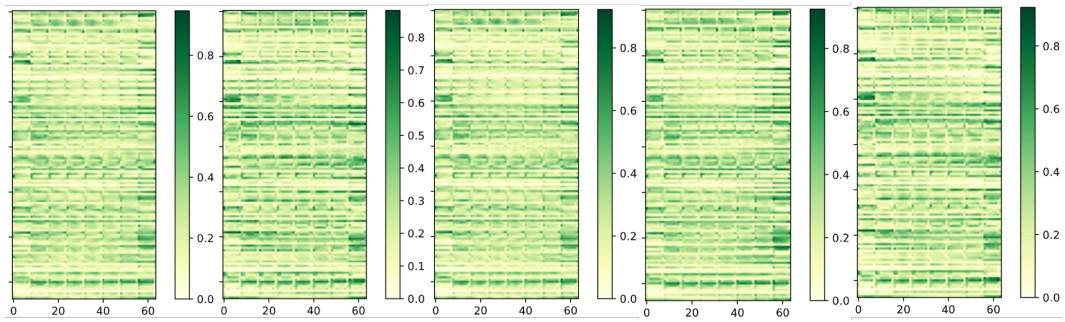

Figure 17: Representation on each task's data with the model learned the last task.

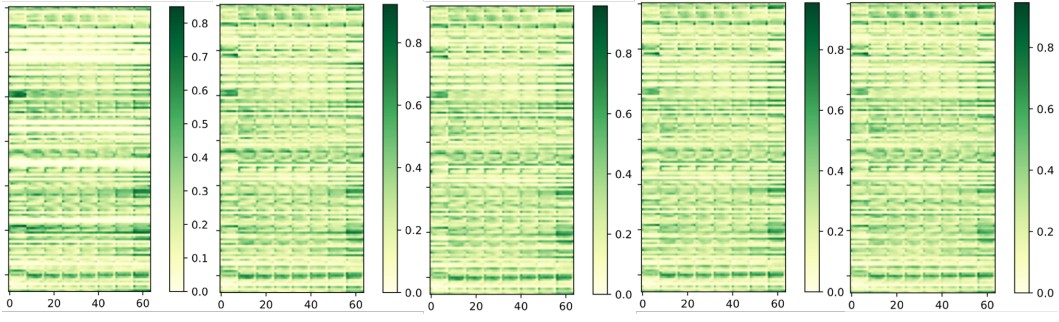

Figure 18: Representation on Task 1 data as learning step progresses.

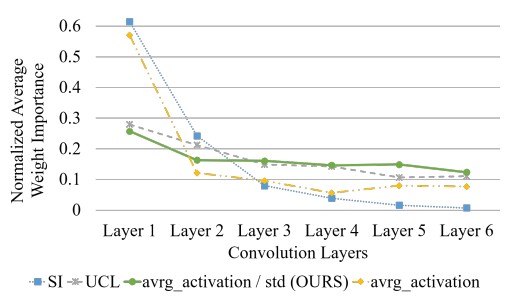

(a) Normalized average of weight importance distribution

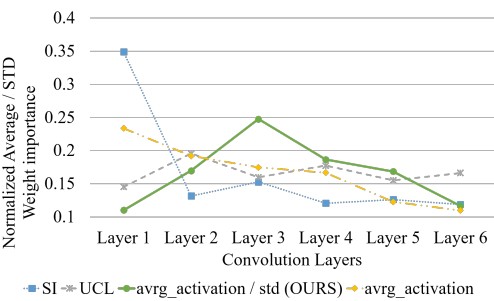

(b) Normalized Average / STD of weight importance distribution

Figure 19: Normalized Weight importance distribution of each convolution layer. 19a represents normalized average of weight importance of each convolution layer. 19b indicates normalized average/std of weight importance of each convolution layer. Our method relaxes the tendency to consolidate weights of earlier layers. This is based on the first task of split CIFAR 10.(task order: 2-0-1-3-4)

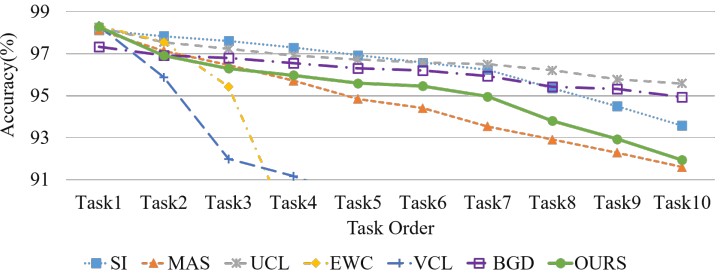

Figure 20: Permuted MNIST Average Accuracy. Due to the lack of memory capacity, we evaluate VCL using multi-layer peceptrons with two hidden layers with 100 ReLU activations.

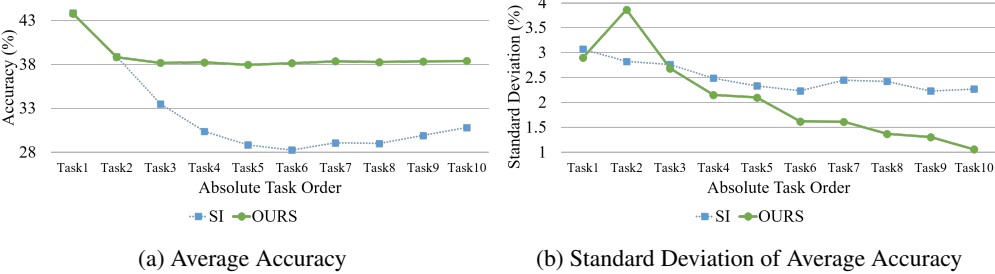

(a) Average Accuracy

(b) Standard Deviation of Average Accuracy

Figure 21: LA accuracy on Tiny ImageNet.

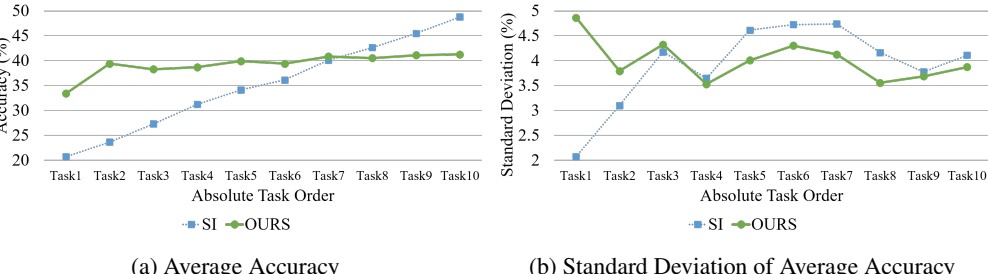

(a) Average Accuracy

(b) Standard Deviation of Average Accuracy

Figure 22: TA accuracy on Tiny ImageNet.

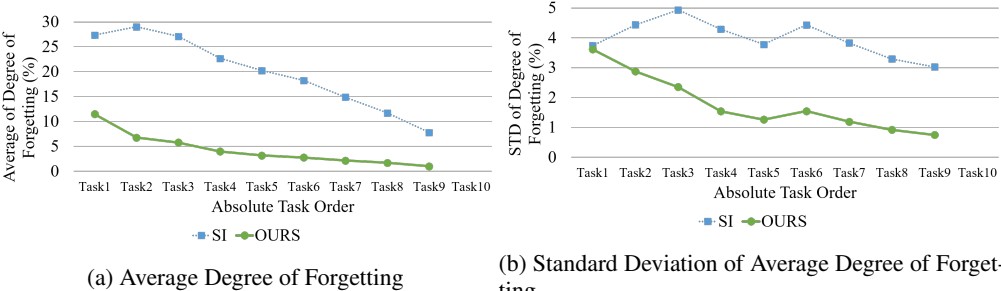

(a) Average Degree of Forgetting

(b) Standard Deviation of Average Degree of Forgetting

Figure 23: Degree of Interference on split Tiny ImageNet. It is calculated by each task's first learned accuracy - accuracy after learning the last task.

