# OpenReview forum: "Continual learning with neural activation importance"
_ICLR.cc/2021/Conference — Reject_

### Official Review · AnonReviewer4 · 2020-10-23
**The paper is rather incremental and does not show its potential sufficiently.**

**Rating:** 4
**Confidence:** 4

**Review:**

This paper introduces a regularization approach for stable continual learning of sequential tasks. The proposed method computes neuron importance based on the activation values of nodes with their respective standard deviation. It further suggests a weight re-initialization scheme to achieve better performance. Experimental results on several continual learning scenarios show that the presented approach is comparable to other existing competitors.

The paper first gives analyses by categorizing existing continual learning strategies based on regularization and tackles an important issue that arises when the order of incoming tasks changes. From the analyses, the paper proposes an approach to defining new neuron importance, as shown in equation (2). However, the presented strategy to find neuron importance is simple and incremental. Moreover, no rigorous analyses of the proposed scheme exist on how it is derived, what are benefits, etc. Thus, I am not convinced that it is particularly useful compared to its strong competitors sharing a similar motivation and strategy. For example, I am not sure why the proposed method can be robust to the order of tasks.

Since the proposed method deals with an order-robust approach, it is compared with other approaches sharing the same goal (e.g., Yoon et al., Scalable and Order-robust Continual Learning with Additive Parameter Decomposition, ICLR 2020). A comparison with existing similar approaches would make this work stronger.

Experimental results do not show the proposed method is promising. The proposed method does not outperform existing works and even performs poorer than some competitors for some experiments. From the results, I am not convinced about the benefits of the proposal. Figure 13 seems not notable output.

---

> ### Author Response · Authors · 2020-11-17
> **Answers to all questions and comments**
>
> The authors appreciate constructive comments. We agree that the novelty of our method is not appropriately explained in the current draft. We have tried to clarify the novelty issue.
>
> Point #1: Novelty and robustness of the proposed method to the order of tasks
>
> Indeed, our regularization term with layer-wise average activation divided by respective std. is a simple extension of prior approaches. This simple modification, however, prevents a particular layer from getting excessive importance compared to other layers, that, in turn, prevent our network from overfitting to a particular task keeping its plasticity regardless of task order.
>
> This is experimentally proved in the following additional simple test. For each method, normalized (average weight importance) of each layer (total 6 layers) is calculated (Figure 19 (a) in our updated draft). Prior average activation based regularization term assigns around 57% of total importance to layer 1 (57%, 12%, 10%, 6%, 8%, 8%, respectively for the 6 layers). On the other hand, our proposed regularization loss term assigns 26% of total importance to layer 1. Furthermore, our method avoids assigning excessive importance to certain layer (26%, 16%, 16%, 15%, 15%, 12%).
>
> Then why this improves the continual learning performance regardless of task order?
>
> In prior work, more weights of lower layers tend to be kept (frozen) in earlier tasks that eliminate the chance of upcoming tasks to build new low-level feature sets. Only a new task that is fortunately able to rebuild higher layer features based on the frozen lower layer weights from previous tasks could survive. On the other hand, ours keeps the balance of frozen weights in all layers securing more freedom of feature descriptions for new tasks in both lower and higher layers.
>
> Indeed, lower layer features such as edges are not class (task) dependent features. Therefore freezing lower layer features actually is not preferable in continual learning. Even though lower level features are mostly changed over tasks, new task may find alternative lower level features keeping its plasticity. On the other hand, if we fail to freeze some higher level features that are class (task) dependent, we may lose robustness. As we have seen, prior works spend most of their importance to lower layers losing its robustness.
>
> We also have compared normalized (average weight importance / standard deviation of the importance of the layer) in Figure 19 (b). Big average activation value means that related weights are critical in the current task for SOME amount of training instances. Small standard deviation of such big average activation value means that the related weights are critical in the current task for almost ALL training instances.
> It clearly shows that our proposed method keeps weights with high popularity allowing importance diversity in each layer. In order to encode such relation, we propose to use average and std. of activation values of each layer rather than using weighs directly. NM-NI type in Figure 1.
>
> To sum up, this test result shows that our regularization term prefers to keep (freeze) weights that have contributed to big average activation with most of training instances. Our method tends to release weights that are critical to relatively small number of training samples.
>
> Point #2: ICLR 2020 paper on the order robustness
>
> Thanks for the information on quite relevant prior work (Yoon et al.). Continual learning of this paper is a type of architecture method as categorized in section 1 of our draft.
> Therefore, directly comparing performance with our regularization method is not appropriate.
> However, Yoon et al. provides nice explanation on the order robustness problem of continual learning and we have introduced the paper in our updated draft.
>
> Point #3: experimental results
>
> Recalling the contribution of our method and new simple test results explained in “Point #1”, most of experimental results on MNIST, Cifar10 and Cifar10-100 that are plotted in the figures in section 3 show that our proposed method tries to keep its performance on all individual tasks.
>
> Accuracy drops of prior work in Figure 5,6,8 show the limited plasticity of them.
>
> On the other hand, accuracy of our method in Figure 5 and 6 does not change too much over tasks.
> Std. of our method in Figure 5 and 6 is lowest indicating that our method is the most robust method to the change of task order.
> Accuracy and std. of our method in Figure 8 show that our method is the most robust method to the change of task order in earlier tasks with cifar10-100 data set as well.
>
> Point #4: Other comments
>
> Fig 13 is removed and other minor comments are all reflected.

---

### Official Review · AnonReviewer3 · 2020-10-28
**Review for Continual learning with neural activation importance**

**Rating:** 4
**Confidence:** 3

**Review:**

This paper proposes a method that tackles the problem of catastrophic forgetting in continual neural networks by assigning importance to neuron activations while tasks are executed in sequence. Similar to previous research (Jung et al., 2020), the proposed method measures neuron importance using average activation values divided by corresponding standard deviation. This strategy is accompanied by weight re-initialization to guarantee that new tasks are fully learned. The method is tested in benchmark datasets for continual learning.

As positive aspects of the paper I would remark:
-	The paper is clearly placed within the existing literature in continual learning, as part of regularization-based approaches.
-	The experimental evaluation tests several options of tasks sequences to prevent results to be dependent on the task order. This is an important aspect to demonstrate the validity of the results.

Weaknesses of the paper are:
-	The main weakness in my opinion is that the proposed method represents a simple extension of previous work (Jung et al., 2020). This compromises the novelty of the proposed approach. Furthermore, the experimental results do not denote a remarkable advantage of the proposed method compared to existing approaches.
-	I am missing some insights on how the experimental results are connected to the motivation of the proposed method. For example, how adding a sense of the standard deviation of neurons importance help to prevent keeping weights of earlier layers in the experimental datasets?
-	The experimental setup is limited to a few tasks, which undermines the results presented in the sense that continual learning systems may be composed of several tasks (hundreds, thousands) received in sequence.

Some remaining comments/questions:
-	There are some minor typos/formatting problems along the paper. E.g. caption in Figure 3 does not mention from which dataset these results come from. The presentation of plots of standard deviations alongside mean accuracies is unnatural – why not to use error bars so the differences are clearer?
-	How adding a sense of the standard deviation of neurons importance help to prevent keeping weights of earlier layers in the experimental datasets?

---

> ### Author Response · Authors · 2020-11-17
> **Answers to all questions and comments**
>
> The authors appreciate constructive comments. We agree that the novelty of our method is not appropriately explained in the current draft. We have tried to clarify the novelty issue.
>
> Point #1: Novelty
>
> Indeed, our regularization term with layer-wise average activation divided by respective std. is a simple extension of prior approaches. This simple modification, however, prevents a particular layer from getting excessive importance compared to other layers, that, in turn, prevent our network from overfitting to a particular task keeping its plasticity regardless of task order.
>
> This is experimentally proved in the following additional simple test. For each method, normalized (average weight importance) of each layer (total 6 layers) is calculated (Figure 19 (a) in our updated draft). Prior average activation based regularization term assigns around 57% of total importance to layer 1 (57%, 12%, 10%, 6%, 8%, 8%, respectively for the 6 layers). On the other hand, our proposed regularization loss term assigns 26% of total importance to layer 1. Furthermore, our method avoids assigning excessive importance to certain layer (26%, 16%, 16%, 15%, 15%, 12%).
>
> Then why this improves the continual learning performance regardless of task order?
>
> In prior work, more weights of lower layers tend to be kept (frozen) in earlier tasks that eliminate the chance of upcoming tasks to build new low-level feature sets. Only a new task that is fortunately able to rebuild higher layer features based on the frozen lower layer weights from previous tasks could survive. On the other hand, ours keeps the balance of frozen weights in all layers securing more freedom of feature descriptions for new tasks in both lower and higher layers.
> Indeed, lower layer features such as edges are not class (task) dependent features. Therefore freezing lower layer features actually is not preferable in continual learning. Even though lower level features are mostly changed over tasks, new task may find alternative lower level features keeping its plasticity. On the other hand, if we fail to freeze some higher level features that are class (task) dependent, we may lose robustness. As we have seen, prior works spend most of their importance to lower layers losing its robustness.
>
> We also have compared normalized (average weight importance / standard deviation of the importance of the layer) in Figure 19 (b). Big average activation value means that related weights are critical in the current task for SOME amount of training instances. Small standard deviation of such big average activation value means that the related weights are critical in the current task for almost ALL training instances.
> It clearly shows that our proposed method keeps weights with high popularity allowing importance diversity in each layer. In order to encode such relation, we propose to use average and std. of activation values of each layer rather than using weighs directly. NM-NI type in Figure 1.
>
> To sum up, this test result shows that our regularization term prefers to keep (freeze) weights that have contributed to big average activation with most of training instances. Our method tends to release weights that are critical to relatively small number of training samples.
>
> Point #2: experimental results
>
> Recalling the contribution of our method and new simple test results explained in “Point #1”, most of experimental results on MNIST, Cifar10 and Cifar10-100 that are plotted in the figures in section 3 show that our proposed method tries to keep its performance on all individual tasks.
>
> On the other hand, accuracy of our method in Figure 5 and 6 does not change too much over tasks. Std. of our method in Figure 5 and 6 is lowest indicating that our method is the most robust method to the change of task order. Accuracy and std. of our method in Figure 8 show that our method is the most robust method to the change of task order in earlier tasks with cifar10-100 data set as well.
>
> Point #3: Limitation to a few tasks
>
> Once we fix network structure, network capacity is also limited allowing limited number of new tasks in continual learning. There exists another stream of continual learning research such as Yoon et al.(2019) where weight decomposition and scalable network scale are employed to address the problem (see sectinon1 of our draft). This is out of the scope of the contribution of our method. We believe that such type of improvement is able to be combined with our new regularization loss as well.
>
> Point #4: Minor comments
>
> - Error bar type average and std. are drawn in Fig 16. It clearly shows std. over average, however, it is little bit difficult to compare avg and std. changes over tasks. We will try to find better representation.
> - Typos and all minor suggestions are all reflected in our updated draft.

---

### Official Review · AnonReviewer1 · 2020-10-28
**Interesting direction, but unclear if there's any meaningful contribution over previous work (Jung et al, 2020)**

**Rating:** 4
**Confidence:** 4

**Review:**

This paper describes an approach to regularization-based continual learning that looks to preserve parameters based on node importance rather than weight importance. The paper categorises existing techniques into these groups (most prior work is based on weight importance), argues that node importance is a better measure, and then performs evaluation on numerous benchmarks and with different variations (importance calculation, task ordering, etc).

This is an important and promising research direction, but unfortunately I think the paper has a number of problems in its current form which preclude publication at this stage.

First, the approach seems to derive extensively from Jung et al (2020), and it is unclear whether there is any algorithmic or technical innovation that is added here. From Section 2.1, the only potential improvement I could see is that the average activation is scaled by the standard deviation, which is quite a trivial change. Reinitialization of weights is also discussed, but this is present in previous work (e.g. Ahn et al (2019)) as well. Figure 3 doesn’t show much of a difference in the distribution of weight importances (beyond the obvious scaling), and most crucially, the experiments do not compare to Jung et al, which makes it very difficult to gauge whether there is any novelty at all.

Second, the approach compares against SI/MAS/EWC as the canonical weight-importance-based approaches, but these are quite old, and a number of more recent techniques also fall into this category, such as VCL [1] (without coreset) and BGD [2] - which estimate the posterior distribution over weights at each timestep, hence determining which ones are important. The experiments would be more convincing if they demonstrated improvement over these more recent approaches.

Last, the writing is quite unclear in parts, and I’d recommend a thorough proofread for spelling/grammar and clarity. A few things I spotted are listed below, but there are many others.

Other questions and comments:
- Clarification for page 1: EWC computes weight importance after each task, not “after network training”
- Not clear why Fig 1b and 1c are necessarily different categories - the only way they are different is how the node importance is calculated. Is this fundamentally a different case, and why?
- One claim in page 3: “If average activation value is used as neuron importance, method will prefer to keep the weights of earlier layers” is stated as a problem - why is this so? This seems reasonable, as lower layers (edge filters, etc) are more likely to be relevant to many tasks, and the higher layers that model more complex features are more likely to need greater plasticity to adapt to new tasks.
- At the start of page 2, it says that with weight-importance based methods, “it is impossible to reinitialize weights at each training of a new task, which decreases the plasticity of the network.” We usually don’t want to reinit weights for each task, so it’d be good to provide more context upfront for why we might want to do this (this comes later in section 2.1.1).

Some typos and writing issues:
- Page 1, not sure what “utilize the weights of a given network to the hit” means.
- Page 2, “One of our key observation…” should be observations; “We propose comprehensive evaluation …” should be “... a comprehensive…”.
- Page 3: “As exampled in …”; “not only networks have to…”; “we can let the model starts…”, etc.

[1] Nguyen, Cuong V., et al. "Variational continual learning." arXiv preprint arXiv:1710.10628 (2017).
[2] Zeno, Chen, et al. "Task agnostic continual learning using online variational bayes." arXiv preprint arXiv:1803.10123 (2018).


=================
Post-discussion:

After reading the authors' response and the changes to the paper, I must unfortunately stick with my current score. I applaud the authors for taking my feedback on board, and certainly many changes have been made to improve the paper, but my main concern of novelty regrettably still remains. The paper offers a very simple improvement over Jung et al (effectively scaling the importance measure), which I believe is quite incremental in this setting. While I believe simple advances can often have broad impact (eg. dropout, batchnorm, etc), in this case it is not clear that the proposed change offers any benefits outside of the very specific area of importance-based continual learning.

---

> ### Author Response · Authors · 2020-11-17
> **Answers to all questions and comments**
>
> The authors appreciate constructive comments. We have tried to clarify all questions and comments.
>
> Point #1: Novelty
>
> Indeed, our regularization term with layer-wise average activation divided by respective std. is a simple extension of prior approaches. This simple modification, however, prevents a particular layer from getting excessive importance compared to other layers, that, in turn, prevent our network from overfitting to a particular task keeping its plasticity regardless of task order.
>
> This is experimentally proved in the following additional simple test. For each method, normalized (average weight importance) of each layer (total 6 layers) is calculated (Figure 19 (a) in our updated draft). Prior average activation based regularization term assigns around 57% of total importance to layer 1 (57%, 12%, 10%, 6%, 8%, 8%, respectively for the 6 layers). On the other hand, our proposed regularization loss term assigns 26% of total importance to layer 1. Furthermore, our method avoids assigning excessive importance to certain layer (26%, 16%, 16%, 15%, 15%, 12%).
>
> Then why this improves the continual learning performance regardless of task order?
>
> In prior work, more weights of lower layers tend to be kept (frozen) in earlier tasks that eliminate the chance of upcoming tasks to build new low-level feature sets. On the other hand, ours keeps the balance of frozen weights in all layers securing more freedom of feature descriptions for new tasks in both lower and higher layers.
> We also have compared normalized (average weight importance / standard deviation of the importance of the layer) in Figure 19 (b). Big average activation value means that related weights are critical in the current task for SOME amount of training instances. Small standard deviation of such big average activation value means that the related weights are critical in the current task for almost ALL training instances.
>
> It clearly shows that our proposed method keeps weights with high popularity allowing importance diversity in each layer. In order to encode such relation, we propose to use average and std. of activation values of each layer rather than using weighs directly. NM-NI type in Figure 1.
>
> Point #2: Reinitialization of weights is already discussed. (Ahn et al(2019))
>
> Jung et al.(2020) mentions weight reinitialization is used for plasticity without any detailed discussion. We agree that the weight reinitialization helps continual learning be optimal in classification keeping its plasticity and we expected to discuss about this more in detail in section 2.1.1.
>
> Point #3: Comparison with Jung et al(2020)
>
> Unfortunately, code for Jung et al.(2020) is not available. Major contribution of Jung et al. includes node importance with average activation value and proximal gradient descent for optimization.
> Our experimental evaluation with (average activation) loss vs (average activation)/(std) loss provides the performance comparison between the two different loss terms. All other aspects of Jung et al. may apply to our method as well.
>
> Point #4: SI / MAS / EWC + VCL / BGD
>
> As the reviewer suggested, we also added VCL (2017), BGD(2018) experimental results together with EWC(2017), SI(2017), MAS(2018). VCL and BGD are, actually, Bayesian neural network based approaches. Figure 4 and 20 show experimental results including VCL and BGD where our proposed method still outperforms all prior approaches.
>
> Point #5: Difference between figure 1b and figure 1c
>
> Yes, difference between “1b” and “1c” are how node importance is calculated. “1b” explicitly considers all connected weights using weight-wise measurement on the importance. Therefore, in this case, node importance is simply the integration of all connected weight importance. On the other hand, “1c” does not have to investigate each connected weight and their importance. It only concerns if current node is important. For example, in case “1c”, a node with several connected weights could be important even if only one of those weight keep sending very high activation, which could be considered as not so important in case “1b” due to very low importance of all other weights.
>
> Point #6: Lower layers (edge filters, etc)
>
> Indeed, lower layer features such as edges are not class (task) dependent features. Therefore freezing lower layer features actually is not preferable in continual learning. Even though lower level features are mostly changed over tasks, new task may find alternative lower level features keeping its plasticity.
> On the other hand, if we fail to freeze some higher level features that are class (task) dependent, we may lose robustness. As we have shown in “Point #1”, prior work spend most of their importance to lower layers losing its robustness.
>
> Point #7: minor comments
>
> All minor typos and suggestions are reflected in our updated draft.

---

### Official Review · AnonReviewer2 · 2020-10-29
**Interesting work but lacks sufficient methodological contributions**

**Rating:** 6
**Confidence:** 4

**Review:**

Summary:

This work focuses on improving continual learning framework by introducing neural importance determined by activation value. In doing so, the authors introduce neuron importance as weight factor in minimizing catastrophic forgetting via regularization term. They also investigate continual learning by changing the order of the tasks.


Strengths:
- introduces the neuron importance in regularization terms for minimizing catastrophic forgetting
- the paper is well written
- sound quantitative evaluation and performance analysis  using several data sets

Weaknesses:
- lack novelty and limited methodological contributions
- it is unclear the need for introducing neuron importance in regularization techniques for catastrophic forgetting, since the activation value of neurons is also derived from the network weights
- the gains are marginal for some data sets


Questions:
- Since neural networks (neuron activity) is too sensitive with change in representation (or input) and therefore, does not guarantee stable results on a particular historical task. This work focuses on determining neuron importance by activation value, however does this activation value correlates to task performance?
- Since neural activity/importance is calculated based on the network weights, introducing neural importance as well as weight regularization is redundant? Please clarify.

Additional comments:
- Unclear, what is f_n_k in equation 2?

Additional references (On Page 1, para 2 or 3):
1. Kirkpatrick, J., Pascanu, R., Rabinowitz, N., Veness, J., Desjardins, G., Rusu, A. A., Milan, K., Quan, J., Ramalho, T., Grabska-Barwinska, A., et al. Overcoming catastrophic forgetting in neural networks. Proceedings of the national academy of sciences  2017.
2. Gupta, P., Chaudhary, Y., Runkler, T., Schütze, H. Neural Topic Modeling with Continual Lifelong Learning. In ICML 2020.

---

> ### Author Response · Authors · 2020-11-17
> **Answers to all questions and comments**
>
> The authors appreciate constructive comments. As the reviewer pointed out, we have introduced an activation value based continual learning to minimize catastrophic forgetting regardless of the order of tasks.
>
> Point #1: Novelty
>
> We agree that the novelty of our method is not appropriately explained in the current draft. Here, we are providing more detailed explanation on the novelty of our work with supporting simple test result (updated in the draft, Figure 19).
>
> Indeed, our regularization term with layer-wise average activation divided by respective std. is a simple extension of prior approaches. This simple modification, however, prevents a particular layer from getting excessive importance compared to other layers, that, in turn, prevent our network from overfitting to a particular task keeping its plasticity regardless of task order.
>
> This is experimentally proved in the following additional simple test. For each method, normalized (average weight importance) of each layer (total 6 layers) is calculated (Figure 19 (a) in our updated draft). Prior average activation based regularization term assigns around 57% of total importance to layer 1 (57%, 12%, 10%, 6%, 8%, 8%, respectively for the 6 layers). On the other hand, our proposed regularization loss term assigns 26% of total importance to layer 1. Our method avoids assigning excessive importance to certain layer (26%, 16%, 16%, 15%, 15%, 12%).
>
> Then why this improves the continual learning performance regardless of task order?
>
> In prior work, more weights of lower layers tend to be kept (frozen) in earlier tasks that eliminate the chance of upcoming tasks to build new low-level feature sets. Only a new task that is fortunately able to rebuild higher layer features based on the frozen lower layer weights from previous tasks could survive. On the other hand, ours keeps the balance of frozen weights in all layers securing more freedom of feature descriptions for new tasks in both lower and higher layers.
> Indeed, lower layer features such as edges are not class (task) dependent features. Therefore freezing lower layer features actually is not preferable in continual learning. Even though lower level features are mostly changed over tasks, new task may find alternative lower level features keeping its plasticity. On the other hand, if we fail to freeze some higher level features that are class (task) dependent, we may lose robustness. As we have seen, prior works spend most of their importance to lower layers losing its robustness.
>
> We also have compared normalized (average weight importance / standard deviation of the importance of the layer) in Figure 19 (b). Big average activation value means that related weights are critical in the current task for SOME amount of training instances. Small standard deviation of such big average activation value means that the related weights are critical in the current task for almost ALL training instances.
> It clearly shows that our proposed method keeps weights with high popularity allowing importance diversity in each layer. In order to encode such relation, we propose to use average and std. of activation values of each layer rather than using weighs directly. NM-NI type in Figure 1.
> To sum up, this test result shows that our regularization term prefers to keep (freeze) weights that have contributed to big average activation with most of training instances. Our method tends to release weights that are critical to relatively small number of training samples.
>
> Point #2: Marginal gains for some data sets.
>
> Yes, our proposed method will not be so effective when there is innate limitation of network capacity. And this is also true to all other methods as long as they use fixed structure for networks in continual learning.
>
> Point #3: Task Performance vs Neuron importance by activation value
>
> As the reviewer exactly pointed out, no neural network guarantees the stability for a particular historical task, and instead, we determine neuron importance by activation value. Of course our new loss term does not necessarily be the optimal in the viewpoint of overall classification performance. We would like to let our network be optimal in the balancing its layer-wise capacity for all sequentially upcoming tasks.
>
> Point #4: Importance by activation vs Importance by weight: are they redundant?
>
> We have explained why we find importance from activation and freeze connected weights in “Point #1: Novelty”. Importance by weight (EM-EI type in Figure 1) and Importance by activation (NM-NI type in Figure 1) have different contributions in continual learning. Using both methods simultaneously may degrade the importance balancing effect of our method.
>
> Point #5: Minor comments
>
> “Unclear, what is f_n_k in equation 2?”  -> We have seen that the notation is misleading and we have revised them all in our updated draft.
> “Additional references”  ->  All suggested new reference papers are included and explained in our new draft.

---

### Decision · Program_Chairs · 2021-01-07
**Final Decision**

**Decision:**

Reject

**Comment:**

Although this paper proposes an intriguing method for using neuron-importance-based regularization to reduce catastrophic forgetting in continual learning, the method is substantially based upon Jung et al (2020), reducing its novelty. Additionally, the experimental evaluation was unconvincing that the proposed method is an improvement over current methods and precisely how the proposed method differs from Jung et al.  The authors are encouraged to revise the paper to incorporate the reviewers suggestions and many of the points the authors raised in their rebuttals, which the reviewers felt were not adequately addressed in the current version of the paper (as mentioned in private discussions among the reviewers).